# Antioxidants Application Enhances Regeneration and Conversion of Date Palm (*Phoenix dactylifera* L.) Somatic Embryos

**DOI:** 10.3390/plants11152023

**Published:** 2022-08-03

**Authors:** Amal F. M. Zein El Din, Rasmia S. S. Darwesh, Mohamed F. M. Ibrahim, Gehan M. Y. Salama, Ibrahim M. Shams El-Din, Walid B. Abdelaal, Ghada A. Ali, Maha S. Elsayed, Ismail A. Ismail, Eldessoky S. Dessoky, Yasmin M. R. Abdellatif

**Affiliations:** 1The Central Laboratory for Date Palm Researches and Development, Agricultural Research Center, Giza 12619, Egypt; amal.zeineldin@arc.sci.eg (A.F.M.Z.E.D.); darweshssrasmia@gmail.com (R.S.S.D.); ibrahim.shams@arc.sci.eg (I.M.S.E.-D.); walid.abdrabo@arc.sci.eg (W.B.A.); ghada_adel_01@hotmail.com (G.A.A.); maha.sobhy@arc.sci.eg (M.S.E.); 2Department of Agricultural Botany, Faculty of Agriculture, Ain Shams University, Cairo 11566, Egypt; dr_yasminmarzouk@agr.asu.edu.eg; 3Botanical Garden Research Department, Horticulture Research institute (HRI). Agricultural ResearchCenter (ARC), Giza 12619, Egypt; genanew2015@gmail.com; 4Department of Biology, College of Science, Taif University, P.O. Box 11099, Taif 21944, Saudi Arabia; i.ismail@tu.edu.sa (I.A.I.); es.dessouky@tu.edu.sa (E.S.D.)

**Keywords:** *Phoenix dactylifera* L., somatic embryos, conversion, polyphenol oxidase, peroxidase, superoxide dismutase

## Abstract

Many embryogenic systems have been designed to generate somatic embryos (SEs) with the morphology, biochemistry, and vigor uniformity of zygotic embryos (ZEs). During the current investigation, several antioxidants were added to the maturation media of the developing somatic embryos of date palm. Explant material was a friable embryogenic callus that was placed in maturation media containing ABA at 0.5 mg L^−1^, 5 g L^−1^ polyethylene glycol, and 10 g L^−1^ phytagel. Furthermore, α-tocopherol or reduced glutathione (GSH) were used separately at (25 and 50 mg L^−1^). These treatments were compared to a widely used date palm combination of reduced ascorbic acid (ASC) and citric acid at 150 and 100 mg L^−1^, respectively, and to the medium free from any antioxidants. The relative growth percentage of embryogenic callus (EC), globularization degree, differentiation%, and SEs number were significantly increased with GSH (50 mg L^−1^). Additionally, the latter treatment significantly enhanced the conversion% of SEs and the number of secondary somatic embryos (SSEs). ASC and citric acid treatment increased leaf length, while α-tochopherol (50 mg L^−1^) elevated the number of leaves plantlet^−1^. GSH at 50 mg L^−1^ catalyzed the activities of polyphenol oxidase (PPO) and peroxidase (POD) in EC and enhanced the accumulation of proteins in SEs.

## 1. Introduction

The date palm (*Phoenix dactylifera* L.) is highly valuable in arid and hot regions, particularly in the Middle East and in Southern Mediterranean countries [1]. For date palms, a typical somatic embryogenesis technique includes many stages such as callus induction, embryogenic callus multiplication, somatic embryos development, and germination [2]. Several studies have been conducted in order to increase the productivity of somatic embryos with high qualitative and quantitative characteristics. Furthermore, these studies were carried out by analyzing the biochemical components of zygotic embryos (ZEs) and simulating their endogenous hormonal, nutritional, and reduced-oxidized environments, as well as providing the appropriate physical conditions throughout somatic embryogenesis [3,4,5]. In general, during growth and development, the cellular redox milieu is altered, and it is detected by special sensors that activate particular metabolic pathways, causing short- and long-term reactions [6]. Cellular antioxidants have been proven in a number of plants to have a strong favorable effect on embryo development. Glutathione (GSH) and glutathione disulfide (GSSG), as well as ascorbic acid (ASC) and dehydroascorbate (DHA), can switch between reduced and oxidized forms. Moreover, early embryonic stages, high ASC/DHA and GSH/GSSG ratios stimulate cellular proliferation. While low ASC/DHA and GSH/GSSG ratios promote embryo development and increase embryos number and conversion rate that improve the quality of produced SEs [4,5,6,7,8,9,10,11].

On the other hand, tocopherols are lipophilic antioxidants produced only by photosynthetic organisms and accumulate in photosynthetic tissue and seeds [12]. Furthermore, tocopherols are a good H donor in the polyunsaturated fatty acid-rich membrane of chloroplasts. Tochopherols play fundamental roles during germination and early seedling development [13].

Generally, auxin, specifically 2,4-D, is known to be necessary for the formation of small embryogenic clusters, and it inhibits somatic embryos from developing further [14]. 2,4-D is a free radical producer, and its concentration in callus induction media has a major impact on subsequent somatic embryo development at the SEs differentiation stage [15]. Furthermore, adding ABA and PEG to the maturation media greatly enhance embryo maturation, but may result in the production of O_2_^−^ and the activation of antioxidant genes [16].

Reactive oxygen species (ROS) intermediates such as superoxide radicals (O_2_^−^), hydrogen peroxide (H_2_O_2_), and hydroxyl radicals (OH^−^) cause membrane lipid peroxidation, protein denaturation, and DNA mutation [17].

Antioxidants include reduced ascorbate (ASC) and reduced glutathione (GSH) as well as ROS-detoxifying enzymes such as superoxide dismutase (SOD) and peroxidases (POD), rapidly detoxify specific ROS [18]. Furthermore, ROS levels must be tightly regulated since an excessive accumulation might cause cell death. Beside their fundamental roles in enhancement of somatic embryogenesis, antioxidants appear to be essential for plants’ oxidative damage protection [6,19].

This study aimed to improve both the quality and quantity of date palm somatic embryos by adding antioxidants to the maturation media (ascorbate, α-tocopherol and reduced glutathione). In addition to comparing protein level and antioxidant enzyme activities within competent embryogenic callus (EC) and within fully developed SEs that subsequently formed during the maturation period.

## 2. Results

### 2.1. Maturation of Embryogenic Callus

#### 2.1.1. Morphological Observations

Relative growth percentage (RG%) of embryogenic callus (EC) increased significantly into 195.6% when GSH was added to the culture medium at 50 mg L^−1^ and followed by GSH at 25 mg L^−1^ and ASC and citric treatment (188.8 and 160.8, respectively). Meanwhile, RG% reached its lowest level (84.6%) when EC was placed on the culture medium devoid of antioxidants (Figure 1A).

When GSH was added to the maturation medium at 50, 25 mg L^−1^ and tocopherol at 50 mg L^−1^, globularization degree considerably improved (4.8, 4.6 and 4.4 degrees, respectively) as shown in Figure 1B.

Differentiation% of the early stage of somatic embryos (Figure 1C) was enhanced greatly when GSH was added to the maturation medium at 50 and 25 mg L^−1^ (72 and 66%, respectively). However, differentiation% of the early stage somatic embryos decreased to its lowest level (39%) in the control treatment.

As illustrated in Figure 1D, the number of somatic embryos jar^−1^ increased (25 embryos jar^−1^) when GSH was added to the maturation medium at 50 mg L^−1^, compared to the control treatment, which had the lowest significant value of SEs number (10 embryos jar^−1^).

#### 2.1.2. Biochemical Observations

After an eight-week incubation period, biochemical analyses in both of embryogenic callus (EC) and regenerated SEs were conducted. Protein concentration insignificantly increased in the embryogenic callus (EC) that was treated with various antioxidants in comparing with the control medium. When SEs that previously formed in a medium containing GSH at 50 and 25 mg L^−1^, protein concentration rose significantly (233.6 and 114.4 mg 100 g^−1^ FW, respectively) as shown in Figure 2A.

The maximum level of polyphenol oxidase (PPO) specific activity was observed after culturing the EC on maturation medium containing GSH at 50 mg L^−1^ (2430 unit mg^−1^ protein), followed by tocopherol treatment at 50 mg L^−1^ (1966 unit mg^−1^ protein) as shown in Figure 2B. The lowest activity of PPO (1526 unit mg^−1^ protein) was recorded by a competent EC that matured on a medium free from antioxidants.

Concerning generated SEs that matured on culture media supplemented with ascorbic and citric acids treatment at 150 and 100 mg L^−1^, PPO specific activity rose significantly followed by tocopherol at 25 mg L^−1^ (5082 and 5059 unit mg^−1^ protein, respectively). However, SEs that matured on GSH containing medium at 50 mg L^−1^ had the lowest level of PPO specific activity (699 unit mg^−1^ protein).

The results shown in Figure 2C showed the specific activity of peroxidase (POD) in both of competent EC and formed SEs The data clearly revealed that EC developed on maturation medium containing 50 mg L^−1^ GSH had a high level of POD specific activity (18,370 unit mg^−1^ protein), followed by EC developed on maturation medium containing 25 mg L^−1^ GSH (14,758 unit mg^−1^ protein). Furthermore, SEs that were formed on maturation medium supplemented with ASC and citric at (150 and 100 mg L^−1^, respectively) exhibited high level of POD specific activity (56,768 unit mg^−1^ protein) at the end of the maturation stage (8 weeks), followed by SEs that were formed on α-tocopherol at 25 mg L^−1^ (31,779 unit mg^−1^ protein).

Regarding superoxide dismutase (SOD) specific activity, No SOD activity was detected after eight weeks of culturing EC on maturation media enriched with antioxidants as presented in Figure 2D. However, SOD specific activity was only found at a low level (12.18 unit mg^−1^ protein) in SEs that developed on GSH treatment (50 mg L^−1^).

### 2.2. Conversion of Mature Somatic Embryos into Viable Plantlets

Generative SEs were examined for germination and conversion processes when cultured on germination medium contained NAA at 0.1 mg L^−1^ and BA at 0.05 mg L^−1^ incubation period after six weeks.

Morphological observations were estimated as conversion%, leaves and roots number plantlet^−1^, leaf and root length (cm) and secondary somatic embryos number jar^−1^ (SSEs).

The percentages of conversion were improved with all antioxidant treatments (Figure 3A). High percentages were reported when date palm fully developed SEs were generated on media supplemented with GSH at 50 mg L^−1^ or ASC and citric acid at 150 and 100 mg L^−1^ (84 and 78%, respectively). SEs formed in an antioxidant-free medium exhibited a low conversion percent (36%).

Leaf number plantlet^−1^ increased significantly (3.6 leaves plantlet^−1^) when SEs were generated in maturation media containing 50 mg L^−1^ α-tochopherol, followed by plantlets derived from SEs matured on 25 mg L^−1^ GSH (3.2 leaves plantlet^−1^) as shown in Figure 3B.

Leaf length reached its greatest value (4.14 cm) when SEs were developed in a maturation medium containing both ASC and citric acids at 150 and 100 mg L^−1^, respectively, but it fell dramatically (1.5 cm) when SEs were formed in a free antioxidant medium (Figure 3C).

Antioxidant treatments didn’t significantly influence the number of roots plantlet^−1^ (Figure 3D). However, GSH treatment at 50 mg L^−1^ enhanced this parameter (1.8 roots plantlet^−1^). While GSH at 25 mg L^−1^ as well as ASC and citric acid treatment gave similar results (1.6 roots plantlet^−1^).

Maturation media supplemented with different antioxidant treatments significantly increased the elongation of roots (Figure 3E). GSH with a concentration of 50 mg L^−1^ greatly enhanced the root length into 6.36 cm (Figure 4e).

The original SEs when placed in the germination medium supplemented with NAA and BA at 0.1 and 0.05 mg L^−1^ germinated as well as multiplied via SSEs (Figure 3F). Additionally, maturation media with GSH or α-tochopherol at 50 mg L^−1^ resulted in increase the number of SSEs significantly (20 and 18 embryos, respectively).

## 3. Discussion

It is generally understood that the redox state is critical during the proliferation process. As a result of fast cell division and active aerobic metabolism, reactive oxygen species (ROS) are generated in greater quantities. ROS are toxic molecules that must be controlled by both enzymatic and non-enzymatic antioxidant mechanisms. Redox chemicals such as ASC, GSH and α- tocopherol play critical roles in plant growth and development, as well as providing antioxidant protection [13,19].

Concerning the relative growth percentage (RG%) which increased by exogenously supplied GSH and ASC, our presented data are in agreement with those obtained by Belmonte et al. [8], who found that GSH treatment promoted white spruce embryogenic tissue proliferation by a 25% increase in fresh weight due to its enhancement of mitotic activity.

Also, endogenous antioxidant levels were discovered to be higher in proliferative cells than in developing cells [6,9,20]. Furthermore, Zein El Din et al. [5] observed that the reduced forms of ASC and GSH were identified in higher amounts in EC than in the degenerative embryogenic callus (DEC) of date palm cv. Barhee grown in vitro.

Sulfhydryl groups are necessary for cell division. However, GSH and cysteine treatments counteract the GSSH-induced suppression of cell division and proliferative growth [21]. Diaz-Vivancose et al. [22] found that GSH accumulated in the nucleus when most Arabidopsis cells were in the G_1_ phase and may induce dividing state during proliferative growth. GSH has long been known to act as an electron donor for dehydroascorbic acid reductase (DHA), allowing ASC to be regenerated through DHA reduction [18].

In general, cell wall extensibility is required for both dividing and expanding cells. Wall ascorbate and ascorbate oxidase are linked to cell growth and wall metabolism [23].

Globularization degree enhanced greatly when GSH and α-tocopherol were added to the maturation media. When both reduced ASC and GSH were added to the culture media, the growth of early stage somatic embryos of loblolly pine increased [9].

However, during this investigation, when ASC and citric acid were added together to the maturation medium at 150 and 100 mg L^−1^, respectively, the degree of globularization lowered (3.4 degrees). It is probable that this was owing to their non-physiological concentrations. In this regard, Gupta and Datta [24] discovered that high concentrations of antioxidants resulted in clear suppression of somatic embryogenesis. ASC, GSH, and other reducing agents may stimulate growth when applied in concentrations matching physiological redox buffer concentrations at appropriate developmental stages.

It is well known that culturing cells in auxin-containing media and in a reduced-environment state leads to the proliferation growth. When cells are placed in culture media that lack auxin, they develop embryonically, and endogenous free IAA levels are reduced [25]. ASC acts as a cofactor for IAA breakdown enzymes [26]. The proposed auxin-antioxidant relationship was supported by Earnshaw and Johnson [20].

Differentiation of early stages of SEs and SEs numbers were positively affected when antioxidants were added to the culture media while free antioxidant medium resulted in reducing these parameters indicating that this maturation medium wasn’t suitable for further development of EC into SEs.

A specific shift in reduced ASC and GSH toward oxidized DHA and GSSH has long been known to encourage histo-differentiation and post-embryonic development [6,10]. Organized growth of somatic embryos occurs in an oxidized environment [20]. GSSH induces several physiological reactions such as shifts in ASC metabolism, synthesis of ABA and ethylene, as well as changes in storage product deposition patterns [6,27]. ABA is widely known for regulating processes during embryo maturation, resulting in the formation of high-quality SEs with a high desiccation tolerance [28]. Moreover, Ghassemian et al. [29] found that ABA-treated seedlings of *Arabidopsis thaliana* caused the accumulation of antioxidants, particularly α-tocopherol and L-ascorbic acid.

The acquisition of competence, induction and development of SEs are associated with increase in protein content during maturation period. Protein content within EC cells increased when EC was placed in antioxidants containing media (Figure 2A). The importance of protein deposition in EC can be explained by its correlation with higher cell division, and metabolic activities in date palm embryogenic callus [5] and in nodular cluster cultures of *Vriesea reitzii* [30]. Moreover, it has been discovered that proteins were stored in the nucleus of EC at a high level [31].

As previously mentioned, the maturation media contained 0.5 mg L^−1^ ABA, 5 g L^−1^ PEG (4000 MW), and 10 g L^−1^ phytagel according to Zein Eldin and Ibrahim [11]; Von Arnold et al. [32] found that ABA promotes normal development by stimulating reserve substance accumulation and suppressing precocious germination in in vitro somatic embryos (according to Ammirato) [33]. ABA increased the generation of ROS molecules and also promoted the expression of several antioxidant enzyme genes [16,27]. Thus, exogenous application of antioxidants may counteract the negative effect of free radicals.

SEs matured on GSH, especially at 50 mg L^−1^, accumulated a high level of protein (Figure 2A). GSH is implicated in a lot of biological processes, including protein and DNA synthesis, enzyme activity, metabolism, and cell protection [34].

Our findings support those of Stasolla [6], who found that external application of GSH during the early stages of embryogenesis increased cellular proliferation and increased the number of embryos, possibly by inducing the synthesis of nucleotides required for energetic processes and mitotic activity. Adding any of the reduced forms of ASC or GSH to early-stage somatic embryos cultures of loblolly pine and Douglas fir Franco improved embryogenic tissue initiation and embryo growth [4,7]. Storage reserves have long been thought to be a good biochemical marker for high-quality SEs and the most successful SEs-to-plantlets conversion [9,11,35].

Moreover, Eliášová et al. [36] reported that storage proteins had an energetic function during germination, which provide nitrogen for de novo synthesis of proteins in Norway spruce SEs as well as their enzymatic and structural functions.

Furthermore, high storage protein content in SEs is associated with the production of so-called Late Embryogenesis Abundant proteins (LEA), which protect embryos from dehydration [31].

PPO specific activity reached its highest significant level in GSH-treated EC; these obtained data are in accordance with those gained by López et al. [37] who reported that PPO reached its peak of activity at the end of the exponential phase of strawberry callus growth. Its activity then dropped, suggesting that this enzyme is involved in flavanol levels regulation in the callus.

PPO regulates free endogenous IAA by oxidizing phenols to quinones, which then condensed with IAA [38]. Furthermore, the formation of auxin-phenol complexes results in the elimination of free endogenous IAA, which has a beneficial impact on the subsequent development of embryogenic callus [11].

Fully developed SEs regenerated on maturation medium contained ASC and citric acid during the current investigation recorded the highest level of PPO specific activity, followed by α-tochopherol treatment (25 mg L^−1^). Tocopherols play a variety of roles in seed viability, seedling growth, and development. On the other hand, tocopherols have been shown to be incorporated into phospholipid bilayers and to serve as a good H donor in these bilayers. Alpha-tochopherol has been discovered to be a non-enzymatic lipid peroxidation protector. Moreover, during the early stages of germination of soybean, there was a decrease in its endogenous content in isolated axes, which may be due to its consumption as a result of an increase in the generation of ROS as imbibition proceeded [39].

Concerning POD specific activity, GSH-treated EC exhibited a high level of its activity. A marked increase in POD activity a day before the globular embryoids was observed in pumpkin callus [40]. An increase was observed in GSH peroxidase and guaiacol peroxidase activities during maturation of the *Eleutherococcus senticosus* somatic embryo [41].

Due to EC cells being characterized by their high cell division and aerobic metabolism activities, the production of H_2_O_2_ and IAA degradation processes are triggered during this period [25]. The uncontrolled synthesis of H_2_O_2_ can cause cellular harm [24,42], as well as being the reason for formation of degenerative EC [5]. As a result, downstream signaling processes are initiated, which stimulate genes that code for ROS-scavenging enzymes.

POD plays a vital function during late-stage embryo development by activating the GSH to GSSH reaction. During the proliferative stage, auxin protectors completely prevent this reaction, maintaining the reduced GSH, allowing cells to divide but inhibiting the development of early-stage to late-stage embryos [43]. In this respect, Cordewener et al. [44] revealed that cationic peroxidase izoenzymes can repair tunicamycin-induced somatic embryogenesis in carrot callus cultures. The findings revealed that such inhibition modified the structure of the pro-embryo masses by causing the surface cells to become more vacuolated and larger, and that POD treatment reversed this by preventing the surface cells from expanding and vacuolating.

Furthermore, it is well recognized that variations in endogenous plant hormones are intimately associated with the development of SEs and their conversion into viable plantlets [45]. ASC acts as a cofactor for enzymes involved in the synthesis of ABA, GA, ethylene, and anthocyanin [26]. As a result, the role of ASC in somatic embryogenesis could be related to its role in modulating the amounts of endogenous plant hormones that ensure the development of SEs in a proper sequence.

The high POD activity and globularization degree observed here with EC grown on medium containing 50 mg L^−1^ GSH confirms the earlier results of El Hadrami [2], who found that an increase in date palm embryogenic callus capacity was associated with a rise in protein content and peroxidase activity. Furthermore, methanolic and phenolic extracts of somatic embryos in the globular stage were found to have higher antioxidant activity than extracts from degenerative embryogenic callus [5].

Regarding SOD specific activity, competent EC didn’t record any level of its activity. However, Gupta and Datta [24] found that the activity of SOD gradually increased in the early stages of gladiolus somatic embryogenesis. Also, the authors reported that there was a reduction in SOD activity with further proliferation of SEs. The only explanation for the lacking of this enzyme, after eight weeks of EC treated by antioxidants, is that the time was late and the enzyme may be expressed earlier at the beginning of callus growth. It has been discovered that SOD acts as the first line defense against harmful effects of ROS molecules.

While SOD specific activity was only discovered when SEs were formed in a maturation medium containing 50 mg L^−1^ GSH, SOD was not detected when SEs were regenerated in other maturation media (including the control group).

During the present study, the authors noted that GSH treatment increased the quality of somatic embryos by increasing protein content, conversion%, and leaf expansion.

Data from other crops, such as peach seedlings [17] and tobacco [46], may help to explain the relationship between expanding leaves and low levels of chloroplastic Cu/Zn SODs. According to their findings, early expanding leaves have a high metabolic activity that leads to an increase in ROS production, necessitating the use of antioxidant enzymes to preserve them.

During this study, the formed SEs were cultured on germination medium containing 0.1 mg L^−1^ NAA and 0.05 mg L^−1^ BA to evaluate the effect of antioxidants on their germination and conversion into viable plantlets as well as test their ability to form secondary somatic embryos (SSEs). Leaf number, leaf length, root length, and SSEs number were greatly increased by adding antioxidants to the maturation media. It is well known that the germination of SEs and their conversion into viable plantlets are clearly dependent on the ability of the apical domes of SEs to resume their dividing state and form the whole plantlets [47].

Leaves number enhanced greatly by adding α-tochopherol (especially at 50 mg L^−1^ (Figure 4f)). Generally, tochopherols play fundamental roles during germination and early seedling development. It has been discovered that absence of tochopherols in mutants Arabidopsis plants resulted in a delay of chloroplast development during the first six days of germination, an oxidative damage of poly-unsaturated fatty acids, and other cell components as well as a sharp defects during seedling growth in comparing to the wild-type [13].

Leaf length increased significantly when both ASC and citric acid were added together (Figure 4d). This obtained data insure the previous data of Liu et al. [48] who found that decreasing in endogenous ASC content resulted in poor seed set and growth rate of transgenic rice seedlings such as plant height, leaves, and roots weight. The authors assumed that ASC is important for cell division and cell elongation.

Root length recorded its highest value when GSH added at 50 mg^−1^ l (Figure 4e). This finding is consistent with those of Vernoux et al. [49], who reported that exogenous GSH enhanced root length by 10–11 mm each day throughout the incubation period of Arabidopsis plants. Moreover, the latter authors hypothesized that GSH is required for the activation and maintenance of cell division and cell cycle activity during the postembryonic growth of roots following germination. The exogenous addition of GSH to Arabidopsis culture media promoted root growth by 15–20%, indicating the stimulatory influence by GSH on root growth [50]. In Arabidopsis, Cheng et al. [51] reported that GSH treatment catalyzed IAA, jasmonic acid, and signaling genes. Due to its activation role for dividing cells, it is possible that increasing IAA biosynthesis as a result of GSH treatment is required for cell division during postembryonic growth of roots. Auxin treatment has favorable effects during in vitro rooting of plantlets.

Moreover, GSH metabolism has a major effect on hair growth and root development [52] as shown here in (Figure 4h), especially at 50 mg L^−1^ concentration. It is possible that adding GSH to the maturation medium significantly increased germination rates, or that GSH acts as an electron donor for DHA reductase, regenerating ASC via DHA reduction (glutathione-ascorbate cycle) as described by Noctor and Foyer [18]. Moreover, an increase in DHA levels during late somatic embryogenesis and within zygotic embryos of date palm cv. Sewi could aid in the de novo synthesis of ASC, resulting in the optimal germination mode into viable plantlets [11]. Additionally, ASC and ASC peroxidase activity are critical for germinability. The relation between ASC, H_2_O_2_, ABA catabolism, and GA synthesis has been well described by Liu et al. [48].

Significant differences in the frequency and number of SSEs were detected in the current investigation between the types and concentrations of antioxidants. SSEs proliferation was significantly increased by treatments with 50 mg L^−1^ for both GSH (Figure 4i,j), and α-tochopherol (20 and 18 embryos jar^−1^, respectively), followed by ASC and citric acid treatment (16 embryos jar^−1^). On the other hand, the control medium gave the lowest value of SSEs (10 embryos jar^−1^).

Currently, secondary somatic embryogenesis is known for its great potential for regeneration. When compared to primary somatic embryogenesis, it has some benefits including a rapid rate of multiplication and repeatability [53].

The findings obtained in the present study indicated that GSH formed-SEs showed ideal morphological features, such as vigor, leaf width (un-shown data), and root length, which may be identical to date palm seedlings, especially at 50 mg L^−1^ (Figure 4e). These embryos had a higher conversion% to viable plantlets; better expanding leaves; and approximately three times longer root length than untreated embryos. Furthermore, adding maturation medium with both ASC and citric acid at (150 and 100 mg L^−1^) resulted in the formation of very strong somatic embryos which had a large apical dome which resemble the zygotic ones (Figure 4c), and the derived plantlets had dark green color (Figure 4g). These findings support earlier findings by Stasolla and Yeung [47], which demonstrated that ASC treatment improved the germination of white spruce somatic embryos by expanding apical regions, which resulted in more leaf primordia, longer shoots, and dark green leaves.

## 4. Materials and Methods

The present study was performed in the tissue culture laboratory of the Central Lab. for Date Palm Researches and Development, ARC, Egypt and Plant Physiology Lab. of Agricultural Botany Department, Faculty of Agriculture, Ain Shams University during (2020–2022). Embryogenic callus cultures were induced from date palm shoot tip explants cv. Sewi grown at Giza Governorate. Healthy selected offshoots were prepared for sterilizing by removing the external layers of tough and fibrous leaf bases using hack-saw until the softer cores were exposed. Then, inside a laminar flow cabinet, the shoot tip explants were sterilized by immersion in mercuric chloride solution (0.1%) for 30 min. After that, explants were rinsed three times in sterile distillated water and then soaked in an antioxidant solution including ascorbic acid and citric acid (150 and 100 mg L^−1^, respectively) until culturing on a callogenesis medium. The shoot tip explants were cut longitudinally into four equal segments and inserted in a culture medium contained 10 mg L^−1^ 2, 4-D + 3 mg L^−1^ 2iP to induce callogenesis according to Zein El Din et al. [5]. The cultures were then re-cultured for three to four subcultures in the freshly prepared MS medium [54] with the same composition and concentrations of plant growth regulators (six-week interval). Subsequently, the formed hardened calli were moved to another medium containing (0.1 mg L^−1^ NAA) for two subcultures in order to achieve further development of calli. Maturation process starts with culturing white friable embryogenic calli (0.5 g jar^−1^) on the MS media containing 0.5 mg L^−1^ ABA, 5 g L^−1^ polyethylene glycol (PEG-4000) plus 10 g L^−1^ phytagel, as well as various types and quantities of antioxidants. Treatments for each of α-tocopherol and reduced glutathione were separately used at concentrations 25, and 50 mg L^−1^. These treatments were compared to the mostly used date palm combination of antioxidants (ascorbic and citric acids at 150 and 100 mg L^−1^, respectively) and a control medium devoid of any antioxidants.

The starting and embryogenic callus (EC) cultures were incubated under complete darkness at 27 ± 2 °C. After an eight weeks incubation period, EC became more competent and the fully developed SEs were also formed in the same cultures (the cultures became a mixture of competent EC, early and late stages of somatic embryos that termed asynchronous somatic embryogenesis). The competent EC and some of generated SEs were morphologically estimated and harvested for the biochemical analyses. The other portion of generated somatic embryos (3 SEs jar ^−1^) were placed on germination medium (NAA at 0.1 mg L^−1^ plus 0.05 mg L^−1^ BA) and incubated in the dark for 7–10 days before being transferred to white light using fluorescent lamps, with the photosynthetic photon flux density (PPFO) set at 40 mol m^−2^ S^−1^. The formed original SEs converted into viable plantlets after six weeks incubation period and several morphological parameters were examined.

### 4.1. Morphological Observations

Relative growth percentage (RG%), globularization degree, differentiation% and somatic embryos number were determined after an eight weeks maturation period on both of EC and formed SEs. Furthermore, the conversion%, leaves and roots number plantlet ^−1^ and, leaf and root length (cm) as well as the number of secondary embryos (SSEs) were also estimated after six weeks from the beginning of culturing the original formed SEs on the germination medium. Observations and photomicrographs were carried out using Stereo Microscope with Integrated LED illumination and Digital 3 MP Camera Leica EZ4 D.

Relative growth percentage (RG%) was calculated as follows:RG% = [(Wn−W0)/W0] × 100
where, Wn is the weight of the callus at the end of incubation and W0 is the weight of the callus at the start of the experiment according to Santoso and Thornburg [55]. Globularization degree was estimated visually as scores according to Pottino [56].

### 4.2. Biochemical Analyses

Determination of soluble protein;Soluble protein concentration was estimated to calculate specific activity of enzymes. Protein concentration was quantified in the crude extract by the method of Bradford [57] using bovine serum albumin as a standard and expressed as mg 100 g^−1^ FW.Assay of peroxidase activity (POD);POD (E.C 1.11.1.7) activity in enzyme crude extract was determined as described by Hammer Schmidt et al. [58]. The activity was calculated by monitoring the change in absorbance per minute at 470 nm. The enzyme unit (IU) is equivalent to 0.01 POD min^−1^. The specific activity was expressed as unit mg^−1^ protein.Assay of Polyphenol oxidase (PPO) activity;According to Benjamin and Montgomery [59], the activity of PPO (EC 1.14.18.1) was tested. One unit of PPO activity was determined as the amount of enzyme that induced a 0.001 per minute increase in absorbance at 420 nm. The enzyme activity was measured in unit mg^−1^ protein.Assay of Superoxide dismutase (SOD) activityThe ability of superoxide dismutase (EC 1.15.1.1) to inhibit the photochemical reaction of nitro blue tetrazolium (NBT) at 560 nm was measured using the Beauchamp and Fridovich [60] method. The amount of protein required to inhibit 50% initial decline of NBT under light is one unit of SOD activity. SOD activity was measured in unit mg^−1^ protein.

All biochemical analyses were determined using a spectrophotometer (UV–Vis spectrophotometer UV 9100 B, Lab Tech).

### 4.3. Statistical Analysis

The experiment design was applied as a complete randomized design. Data were introduced as the mean of the five replicates ± standard deviation. Results were statistically analyzed using Costat software [61] according to Snedocor and Cochran [62] and compared with Tukey’s Studentized Range test by the honest significant difference (HSD) at *p* ≤ 0.05 levels between treatments.

## 5. Conclusions

Somatic embryogenesis is the process by which somatic cells are encouraged to become more competent and switch to embryogenic cells. These cells then continue to develop and produce somatic embryos. Antioxidants especially GSH appear to have a vital role in date palm somatic embryogenesis via increasing the proliferative growth which may estimate by increasing relative growth%, as well as enhancing the development of early stage of somatic embryos. Many physiological correlations are highlighted in the current study based on protein accumulation and antioxidant enzyme activities in competent EC and fully developed somatic embryos, which are important in the development of early and late stages of SEs (Figure 5). Finally, an increase in the productivity of date palm SEs with great quantitative and qualitative features is required for the best possible conversion into plantlets.

## Figures and Tables

**Figure 1 plants-11-02023-f001:**
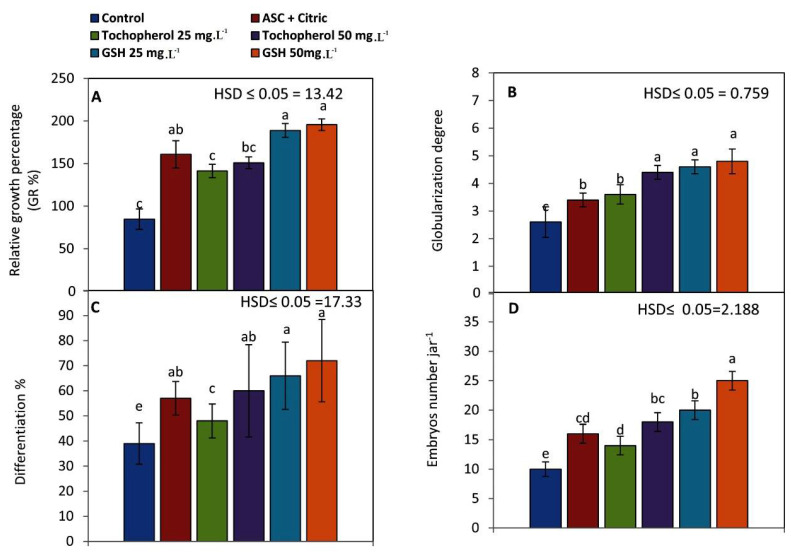
The influence of antioxidants added to date palm maturation media on: (**A**) relative growth% of embryogenic callus, (**B**) globularization degree, (**C**) differentiation% of early stage somatic embryos, and (**D**) SEs number. Bars represent the means of five replicates ± standard deviation; Different letters (a–e) indicate significant differences according to Tukey’s Studentized Range (HSD) test (*p* ≤ 0.05).

**Figure 2 plants-11-02023-f002:**
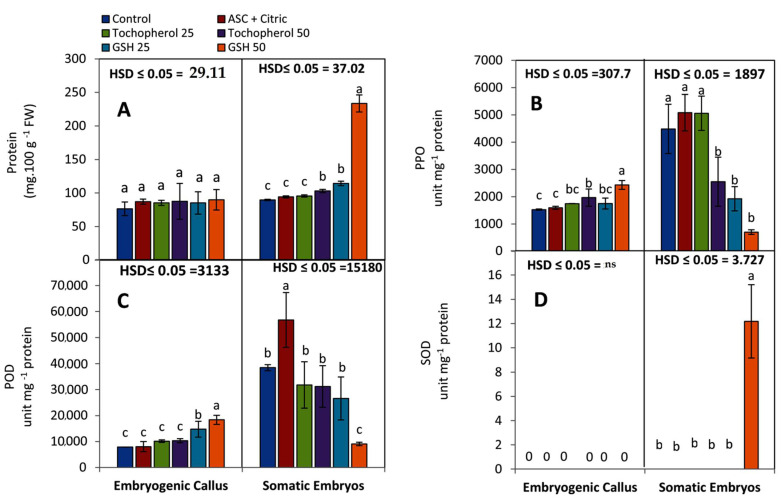
The effect of different antioxidants added to the date palm maturation media on: (**A**) protein concentration (mg/100 g FW), and the specific activities (unit mg^−1^ protein) of (**B**) polyphenol oxidase (PPO), (**C**) peroxidase (POD), and (**D**) superoxide dismutase (SOD) within competent embryogenic callus (EC) and fully developed SEs after eight weeks of treatment. Bars represent the means of five replicates ± standard deviation; Different letters (a–c) indicate significant differences according to Tukey’s Studentized Range (HSD) test (*p* ≤ 0.05).

**Figure 3 plants-11-02023-f003:**
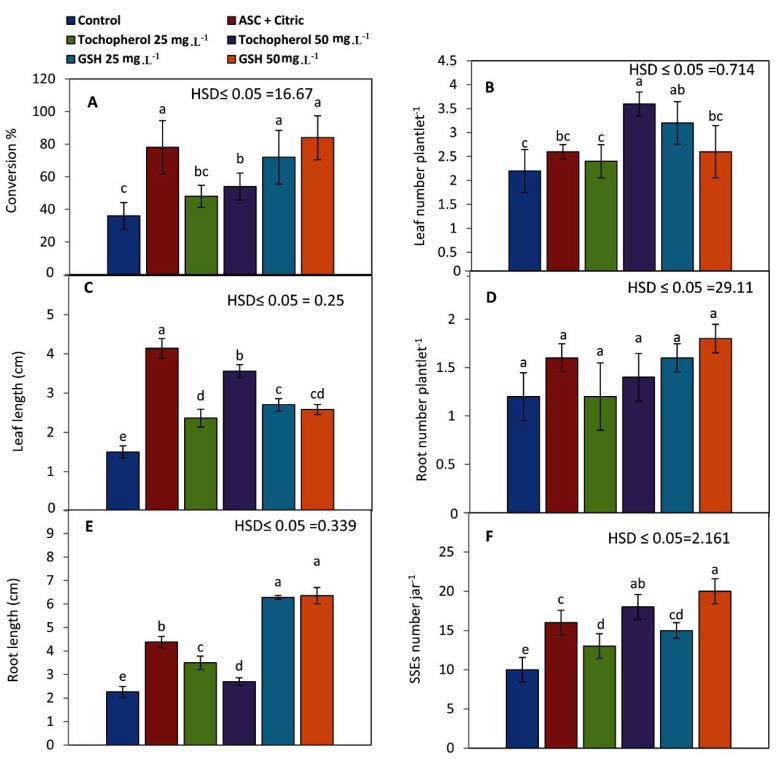
Influence of antioxidants on: (**A**) conversion% of SEs into viable plantlets, (**B**) leaves number plantlets^−1^, (**C**) leaf length (cm), (**D**) root number plantlet^−1^, (**E**) root length (cm), and (**F**) secondary embryos (SSEs) number when regenerated SEs matured previously on different antioxidants treatments and cultured on germination medium (NAA at 0.1 mg L^−1^ plus 0.05 mg L^−1^ BA). Bars represent the means of five replicates ± standard deviation; Different letters (a–e) indicate significant differences according to Tukey’s Studentized Range (HSD) test (*p* ≤ 0.05).

**Figure 4 plants-11-02023-f004:**
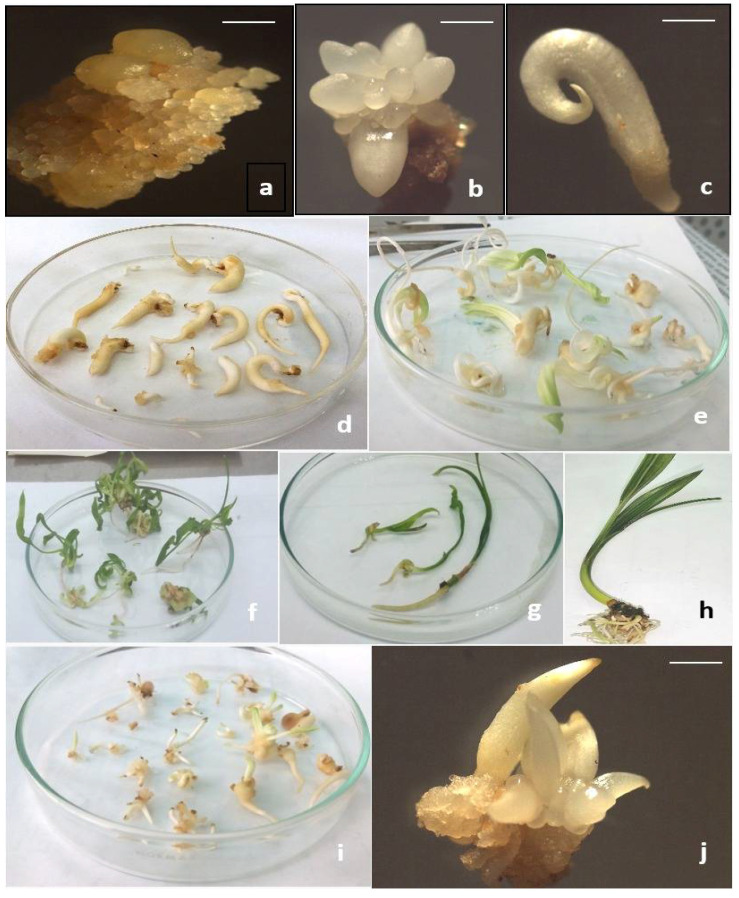
Stages of date palm somatic embryogenesis cultivar Sewi: (**a**) EC, (**b**,**c**) early and late stages of somatic embryos that matured on culture media contained 0.5 mg L^−1^ ABA, 5 g l^−1^ PEG-4000, 10 g l^−1^ phytagel and various types and quantities of examined antioxidants, (**d**) high quality of regenerated SEs treated with ASC and citric acid, (**e**) viable plantlets derived from vigor SEs previously matured on medium supplemented with GSH at 50 mg L^−1^, (**f**) SEs converted into plantlets which characterized with the highest leaves number when maturation medium was supplemented with α- tochopherol at 50 mg L^−1^, (**g**) dark green color of ASC and citric acid treated-plantlets, (**h**) elongation and rooting of plantlets derived from SEs previously matured on GSH at 50 mg L^−1^, and (**i**,**j**) highest number of secondary somatic embryos (SSEs) when original SEs were matured on GSH at 50 mg L^−1^. Bar; a = 0.1 cm, b = 0.25 cm, c = 0.7 cm and j = 0.5 cm.

**Figure 5 plants-11-02023-f005:**
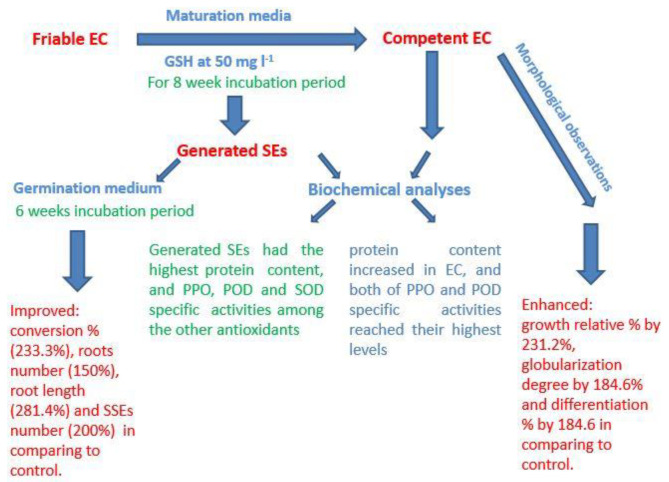
A summary of some morphological and biochemical observations of date palm competent embryogenic callus (EC) and the generated somatic embryos (SEs) as a result of exogenous addition of glutathione (GSH) at 50 mg L^−1^ to the maturation media.

## Data Availability

Not applicable.

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
