# Peer review of "Antioxidants Application Enhances Regeneration and Conversion of Date Palm (Phoenix dactylifera L.) Somatic Embryos"

_plants, 2022, doi:10.3390/plants11152023_

Round 1

Reviewer 1 Report

Review concerns:

Manuscript ID: plants-1825184

Title: Alpha-tocopherol and Reduced Glutathione as Promising Alternative Antioxidants in the Media to Enhance Regeneration and Conversion of Date Palm Somatic Embryos

Authors: Amal F. M. Zein El Din, et al.

The authors claim that Alpha-tocopherol and Reduced Glutathione enhance somatic embryogenesis in date palm. Some cases showed in graphs can confirm this, however, in general the paper seems to be similar to the previous paper with some more biochemical details. Please check this paper:

Zein El Din, A.F.M, et al. Morpho-Anatomical and Biochemical Characterization of Embryogenic and Degenerative Embryogenic Calli of Phoenix dactylifera L. Horticulturae 2021, 7, 393. https://doi.org/10.3390/horticulturae7100393

Neither abstract, nor results convince readers about what authors indicated in the title, lack of coincidence between abstract, introduction. results and final conclusion. In  conclusion section, the authors wrote about final advantage of using GSH- for plantlets and viable plant regeneration. However in the paper I can’t find any data or documentation that confirm that conclusion.

The results are highly unclear with many numbers and graphs but with no clear written and graphic result summary to facilitate reader check this information. The presented photographic documentation is less than acceptable to be published in good scientific journal (and it looks like copied and converted to this manuscript. Please check in the figure 4 abnormally stretched letters and unnatural length plant material, e.g. Fig 4a.

To sum up, the data presented by the authors may be valuable but need deep reediting to be more clear and facilitate readers understand the importance of this research. And the presented data need to be supplemented with the evidence for really better plant regeneration, through the Alpha-tocopherol and Reduced Glutathione usage.

Author Response

Dear reviewer I

We would like to thank you so much for effort, time and your valuable comments which helped us to improve the overall quality of manuscript.

Please find the attached file of our responses to all comments point by point

Thank you so much again 

Warmest regards

The authors

Reviewer 2 Report

This study investigates an important problem with tissue culture in general and not only with date palm cultures. Browning of the developing callus is often seen as a major obstacle for the further development into regenerated plantlets. In this study several antioxidants in different concentration are tested for the influence on the different stages of development from embryogenic callus to plants with roots. Further this is correlated to the content of ex protein in the callus and embryos.

The outcome of what antioxidant and in what concentration give the best “production” of plants is useful for the community especially if this could be correlated with endogenous contents.

For all data the statistics have been performed only the number of explants per replicate is not included in the methods. This is important to be able to evaluate the results of this study. 

Author Response

Dear reviewer II

Thank you so much for time, effort, and your valuable comments which helped us to improve the overall quality of this manuscript

please find the attached file of our responses to all comments point by point

Hopefully, the revised version now meets your high expectations

Warmest regards

The authors

Round 2

Reviewer 1 Report

Review concerns:

Manuscript ID: plants-1825184revised

Thank you the authors for all explanations. I understand the complexity of the phenomenon, especially if a few different exogenous antioxidant were used.

Generally I accept all prepared correction, with one exception - fig. 5. It is a very good idea to prepare that summary figure, however still this figure rather confuses than clarifies the obtained data. “Without antioxidants" mean - control, yes?

So figure should be a simple picture of listed parameters  that increased after antioxidant  using, if possible - how much more, comparing to the control. In current version, readers still don’t know how each antioxidant works (well or not), they can only notice the concentration of used chemicals. So arrows should clear show “increase” and how much (%) in bracket. It would be more better if maybe to choose one, the best antioxidant for this summary and clearly show the advantage of its utility.’

Hope that my comment will help the authors and the readers with better understanding the issue.  

Minor comment:

 Line 109-110 page 3”

2.1.2. Biochemical observations 

After 8-weeks incubation period, Biochemical analyses in both of embryogenic callus (EC) and regenerated SEs – any verb? .. were conducted? Please complete/clarify this sentence.

Author Response

Dear reviewer 1

Thank you so much for you effort and time

We have revised and corrected our MS according to your directions

Please find the attached file of our responses 

Warmest regards

The authors
